# Control Experiences Regarding Clearable Materials from Nuclear Power Plants and Nuclear Installations: Scaling Factors Determination and Measurements' Acceptance Criteria Definition

**Luca Albertone [1],\*, Massimo Altavilla [2], Manuela Marga [1], Laura Porzio [1], Giuseppe Tozzi [1] and Pierangelo Tura [1]**

[1]   Arpa Piemonte, Dipartimento Rischi fisici e tecnologici, Struttura Semplice Radiazioni ionizzanti e Siti Nucleari, Via Trino 89, 13100 Vercelli, Italy; m.marga@arpa.piemonte.it (M.M.); l.porzio@arpa.piemonte.it (L.P.); g.tozzi@arpa.piemonte.it (G.T.); p.tura@arpa.piemonte.it (P.T.)

[2]   ISIN (Ispettorato Nazionale per la Sicurezza Nucleare e la Radioprotezione, National Inspectorate for Nuclear Safety and Radiation Protection), Via Capitan Bavastro 116, 00154 Roma, Italy; massimo.altavilla@isinucleare.it

\*   Correspondence: l.albertone@arpa.piemonte.it

**Abstract:** Arpa Piemonte has been carrying out, for a long time, controls on clearable materials from nuclear power plants to verify compliance with clearance levels set by ISIN (Ispettorato Nazionale per la Sicurezza Nucleare e la Radioprotezione - National Inspectorate for Nuclear Safety and Radiation Protection) in the technical prescriptions attached to the Ministerial Decree decommissioning authorization or into category A source authorization (higher level of associated risk, according to the categorization defined in the Italian Legislative Decree No. 230/95). After the experience undertaken at the "FN" (Fabbricazioni Nucleari) Bosco Marengo nuclear installation, some controls have been conducted at the Trino nuclear power plant "E. Fermi," "LivaNova" nuclear installation based in Saluggia, and "EUREX" (Enriched Uranium Extraction) nuclear installation, also based in Saluggia, according to modalities that envisage, as a final control, the determination of γ-emitting radionuclides through in situ gamma spectrometry measurements. Clearance levels' compliance verification should be performed for all radionuclides potentially present, including those that are not easily measurable (DTM, Difficult To Measure). It is therefore necessary to carry out upstream, based on a representative number of samples, those radionuclides' determination in order to estimate scaling factors (SF), defined through the logarithmic average of the ratios between the i-th DTM radionuclide concentration and the related key nuclide. Specific radiochemistry is used for defining DTMs' concentrations, such as Fe-55, Ni-59, Ni-63, Sr-90, Pu-238, and Pu-239/Pu-240. As a key nuclide, Co-60 was chosen for the activation products (Fe-55, Ni-59, Ni-63) and Cs-137 for fission products (Sr-90) and plutonium (Pu- 238, Pu-239/Pu-240, and Pu-241). The presence of very low radioactivity concentrations, often below the detection limits, can make it difficult to determine the related scaling factors. In this work, the results obtained and measurements' acceptability criteria are presented, defined with ISIN, that can be used for confirming or excluding a radionuclide presence in the process of verifying clearance levels' compliance. They are also exposed to evaluations regarding samples' representativeness chosen for scaling factors' assessment.

**Keywords:** clearance; scaling factor; decommissioning; nuclear power plant; nuclear installation

## 1. Introduction

Until the early 2000s, it was noted that there were, within certain conditions, relatively constant ratios among some radionuclides' activity concentrations for each specific waste stream. For example, the "constant" ratio between the concentrations of Ce-144 and plutonium isotopes in the waste streams generated by a LWR (light water reactor) was proven [1]. The implication of this situation was immediate. It was indeed possible, once the above relationship would have been defined, to evaluate plutonium isotopes' concentrations, which gave evident problems in terms of radiometric measurements, through the easier cerium determination. A large literature is available about this [1–3], essentially based on an American origin which, limited to light water reactors, provides an extensive series of scaling factors (SF) for the different waste currents and different radionuclides of interest. However, using values coming from literature has a limitation due to the fact that there is a wide variability of scaling factors, as well as from waste stream, even from different nuclear installations and from the different way of operating on the same nuclear installation. From this, it follows that scaling factors' values found in the literature could find general applications from which general indications could be extracted. Nevertheless, it was necessary to establish specific evaluations regarding scaling factors, through experimental investigations, when more precise technical information was necessary for defining, for example, a nuclear installation radioactivity inventory.

Nowadays in the international panorama, an average value is used for scaling factors' calculation, in some cases an arithmetic mean, while in other cases, a geometric (logarithmic) average, as in the case of Italy or the United States of America [4]. The concept of scaling factors is based on the assumption that the relationship between a key radionuclide and a radionuclide of difficult measurability is linear in the range of activities of interest (Figure 1).

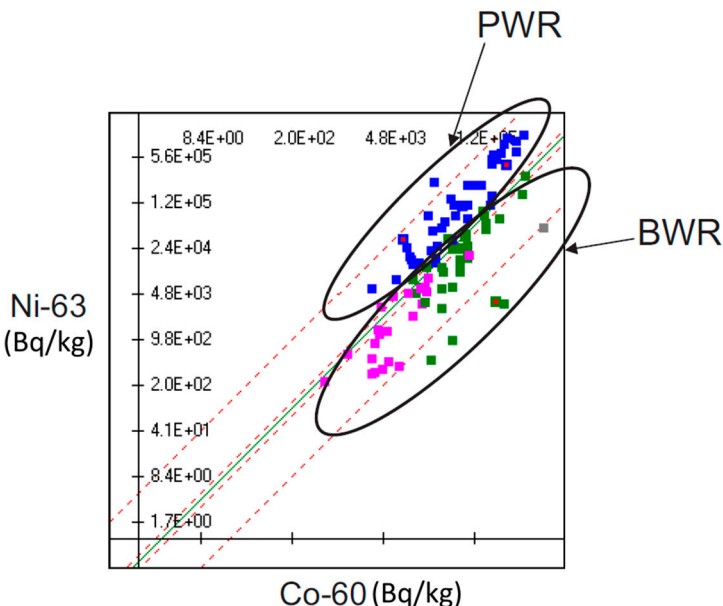

**Figure 1.** Ni-63~Co-60 correlation example for different reactor types: pressurized water reactor (PWR) and boiling water reactor (BWR). Adapted from Determination and Use of Scaling Factors for Waste Characterization in Nuclear Power Plants, © IAEA, 2009 [5].

The arithmetic mean will tend to produce a conservative value while the geometric mean will tend to produce a more representative mean value when the data is distributed over several orders of magnitude. In the international panorama, some countries use logarithmic regression for scaling factors evaluation. In this case, the same correlation factors assume a non-linear relationship between the key nuclide and the difficult to measure (DTM) nuclide, and can be used to more accurately model the complex and non-linear relationships among radionuclides.

## 2. Materials and Methods

In the national context of scaling factors, referred to specific homogeneous groups, are determined through the application of the EPRI (Electric Power Research Institute) [1] method, which recognizes, in the calculation of the geometric mean and the dispersion associated with the distribution of the measured ratios (experimental scaling factors), the most effective formalism in the statistical analysis of experimental data. Based on nuclear installation operational history, on the available radiometric data and on the systems, structures and radioactive waste subdivision in groups such as appropriate homogeneous radiological conditions are suitable, a minimum number of representative materials or waste samples [6] are taken (typically N = 20) for each one of the hypothesized homogeneous groups, so as to compute the i-th experimental scaling factor (SF) for each one of the N samples subjected to radiometric analysis:

$$SF = \frac{C_{DTM}}{C_{ETM}},\tag{1}$$

where, $C_{DTM}$ represents DTM concentration and $C_{ETM}$ represents the easily (Table 1) reference measurable radionuclide concentration (ETM, easy to measure).

**Table 1.** Reference key nuclide for different DTMs.

| DTM Radionuclide | ETM Radionuclide |
|---|---|
| Fe-55, Ni-59, Ni-63 | Co-60 |
| Sr-90, Pu-238, Pu239/240, Pu-241 | Cs-137 |
| Total-U | U-238 |

DTM (difficult to measure), ETM (easy to measure).

The determination of ETM radionuclides is performed by gamma spectrometry—in laboratory [7] and in situ [8], while for the DTM radionuclides, different radiochemical methods are used [9–13].

Afterwards, the geometric mean $A_{SF}$ is calculated as:

$$A_{SF} = e^{\frac{\sum_{i=1}^{N} \ln SF_i}{N}},\tag{2}$$

and the $D_{SF}$ dispersion parameter of the $N$ experimental scaling factors is:

$$D_{SF} = e^{\sqrt{\frac{\sum_{i=1}^{N} (\ln SF_i - \ln A_{SF})^2}{N-1}}}.\tag{3}$$

The $D_{SF}$ definition [1,14] comes from the standard deviation (SD) meaning, applied to the logarithmic measurement distribution $SF_i$. Given a value distribution $\ln(FS_i)$, the probability that a new determination for $\ln(SF)$ gives a result belonging to the range $\ln(A_{SF}) - \ln(D_{SF}) \leq \ln(SF) \leq \ln(A_{SF}) + \ln(D_{SF})$ is 68.3%. Analogue probability, in the corresponding measurement distribution of $SF_i$, results are associated with a new SF measurement whose value is in the range $A_{SF}/D_{SF} \leq SF \leq A_{SF} \cdot D_{SF}$. If $2\sigma = 2\ln(D_{SF})$, to the dispersion of distribution value $\ln(SF_i)$ is associated, when there will be a new $\ln(SF)$ determination, a probability of 95.5% to assume a value in the range $\ln(A_{SF}) - 2\ln(D_{SF}) \leq \ln(SF) \leq \ln(A_{SF}) + 2\ln(D_{SF})$, or $A_{SF}/D_{SF}^2 \leq SF \leq A_{SF} \cdot D_{SF}^2$ for SF measurements.

Scaling factor acceptability coming from the geometric mean of the N experimental scaling factors is therefore established coherently with conservative criteria, indicated by the EPRI [1] and NRC (United States National Regulatory Commission) [15], relative to the $2\sigma$ dispersion ($D_{SF}^2$) of the distribution of experimental scaling factors. According to NRC [15], it can be considered an acceptable target that scaling factors are accurate within a factor of 10, or $D_{SF}^2 \leq 10$.

The nuclear regulatory body, ISIN, on the basis of indications, present in several international publications [1,5,15] and based on the solid materials radiological characterization, in particular relating

to the clearable solid materials, has adopted the criterion $D_{SF}^2 \leq 6$ while, in the case of radioactive wastes radiological characterization, the criterion $D_{SF}^2 \leq 8$ has been adopted.

In the event that the $2\sigma$ dispersion ($D_{SF}^2$) of the distribution of the measurements satisfies the previous requirements, both in the case of clearable materials and radioactive wastes, the statistical consistency of the number of samples subjected to testing would be demonstrated and, consequently, the geometric mean would constitute a reliable estimate of the scaling factor for the generic nuclide representative of the set of materials or waste, constituting the homogeneous group under examination. Conversely, if the aforementioned requirements are not met, the definition of a valid scaling factor should be waived and any other criterion considered to be appropriately conservative should be used.

## 3. Results

This paper does not claim to provide an overview of all the experiences regarding the determination of scaling factors for clearable materials, but only a brief report of the experiences carried out by Arpa Piemonte at:

- Bosco Marengo "FN" nuclear installation;
- Trino "E. Fermi" nuclear power plant;
- Saluggia "LivaNova" nuclear installation;
- Saluggia "EUREX" nuclear installation.

### 3.1. Bosco Marengo "FN" Nuclear Installation Experiences

The Bosco Marengo "FN" nuclear installation carried out its activity in the nuclear fuel cycle field from 1972 to 1990 as the only national nuclear fuel manufacturer for ENEL's (Ente Nazionale per l'Energia Elettrica) nuclear power plants. Currently, following the issue of Ministerial Decree 27 November 2008 for decommissioning authorization, the plant is completing the decommissioning operations, an activity that has involved the production, in addition to radioactive wastes, of considerable quantities of metallic materials, building materials, and various other materials intended for clearance.

The nuclear fuel produced during the operation of the plant consisted of natural and low enriched uranium; therefore, the radionuclides potentially present are: U-238 and U-235 in percentages by mass equal to the enrichment values worked (in the range 0.2–5%) and U-234 in variable proportions depending on the enrichment values.

Following the plant characteristics and the production cycle, the liquid effluents have been considered as characteristic of the average contamination of all the plant components.

Scaling factors were therefore updated [16] through a statistical analysis of the isotopic composition—obtained by alpha spectrometry after radiochemical separation—of the liquid effluents collected during the period 2006–2018. In particular, it was possible to observe an average enrichment of about 2% with an approximately normal distribution. In this case, therefore, there is a statistically significant correlation (Figure 2) between U-238's and total uranium (Total-U) concentrations. It was therefore established to take the U-238 as reference radionuclide (key nuclide).

In the 2013–2017 period, an amount of 12 lots, part of a total of 22 lots, consisting of cleared metallic materials, were subjected to control by Arpa Piemonte using the clearance levels contained in the technical prescriptions of the decommissioning authorization attached to the 27 November, 2008 Ministerial Decree (Tables 2 and 3).

**Table 2.** Metallic materials clearance levels.

| Radionuclide | Metallic Materials | | |
|---|---|---|---|
| | **Direct Reuse** | **Recycle** | **Direct Reuse/Recycle** |
| $\alpha$ emitters | 0.1 Bq/cm$^2$ | 0.1 Bq/cm$^2$ | 1 Bq/g |

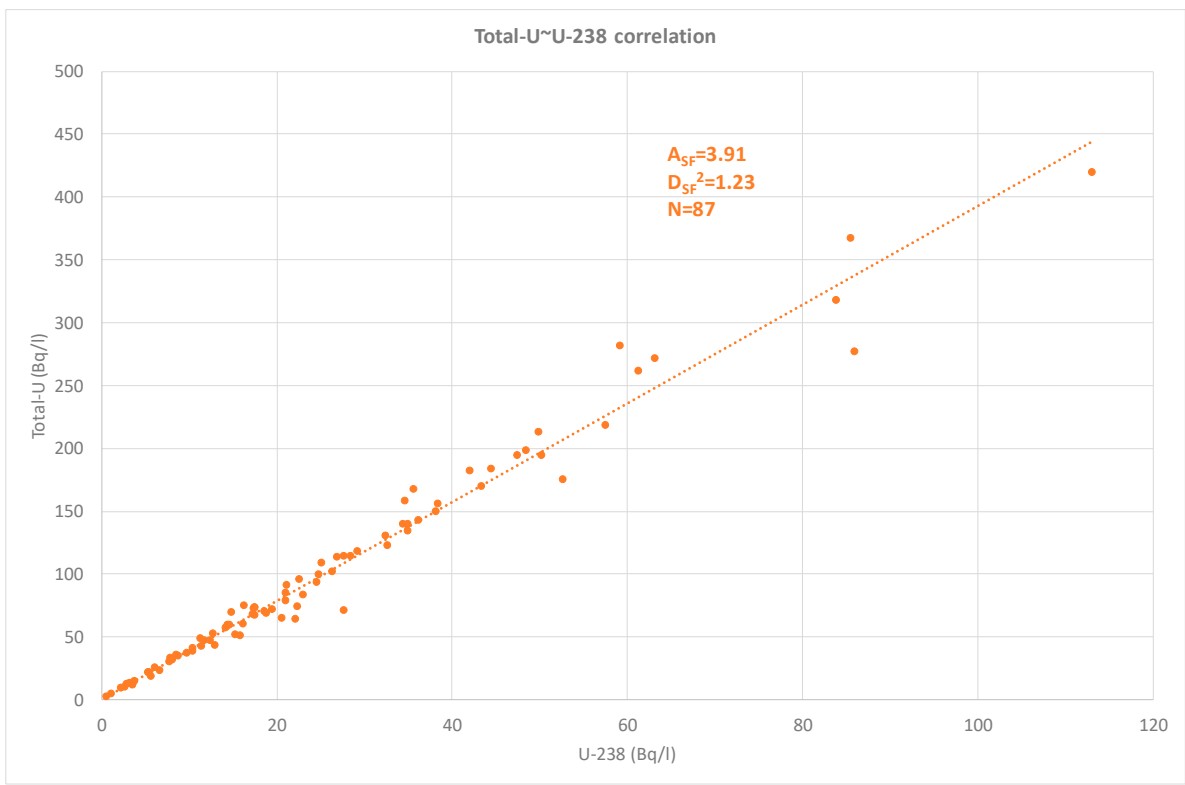

**Figure 2.** Total-U~U-238 correlation in the Bosco Marengo "FN" nuclear installation's liquid effluents. On the x-axis, the concentration of U-238 as determined by alpha spectrometry [10], on the y-axis, the sum of the concentrations of U-238, U-234, and U-235, also determined by alpha spectrometry. The scaling factor $A_{SF}$ is the slope of the line shown in the graph.

**Table 3.** Non-metallic materials clearance levels.

| Radionuclide | Cementitious Materials | | | Various Materials |
|---|---|---|---|---|
| | Buildings Reuse | Demolition | | Mass Concentration |
| | | Surface | Building Rubble | |
| α emitters | 0.1 Bq/cm$^2$ | 1 Bq/cm$^2$ | 0.1 Bq/g | 0.1 Bq/g |
| Other U-238 and U-235 decay products different from Legislative Decree n. 230/1995 Table I-2, Annex I | 0.1 Bq/cm$^2$ | 0.1 Bq/cm$^2$ | 0.1 Bq/g | 0.01 Bq/g |

A total of 376 tons of iron and steel were cleared and all the γ spectrometry measurements confirmed the respect of the clearance levels (Figure 3). U-238 was determined by direct γ spectrometry [8], while Total-U was estimated using the above scaling factor.

*3.2. Trino "E. Fermi" Nuclear Power Plant Experiences*

In the case of Trino "E. Fermi" nuclear power plant, the analysis of the discharges (Figure 4) did not allow us to identify scaling factors because the resulting dispersion did not result to be in compliance with EPRI criteria. This circumstance is not surprising since the effluents derive from a mixing of all the waste streams coming from the different systems of the plant, so that results are hardly traceable to a single homogeneous group.

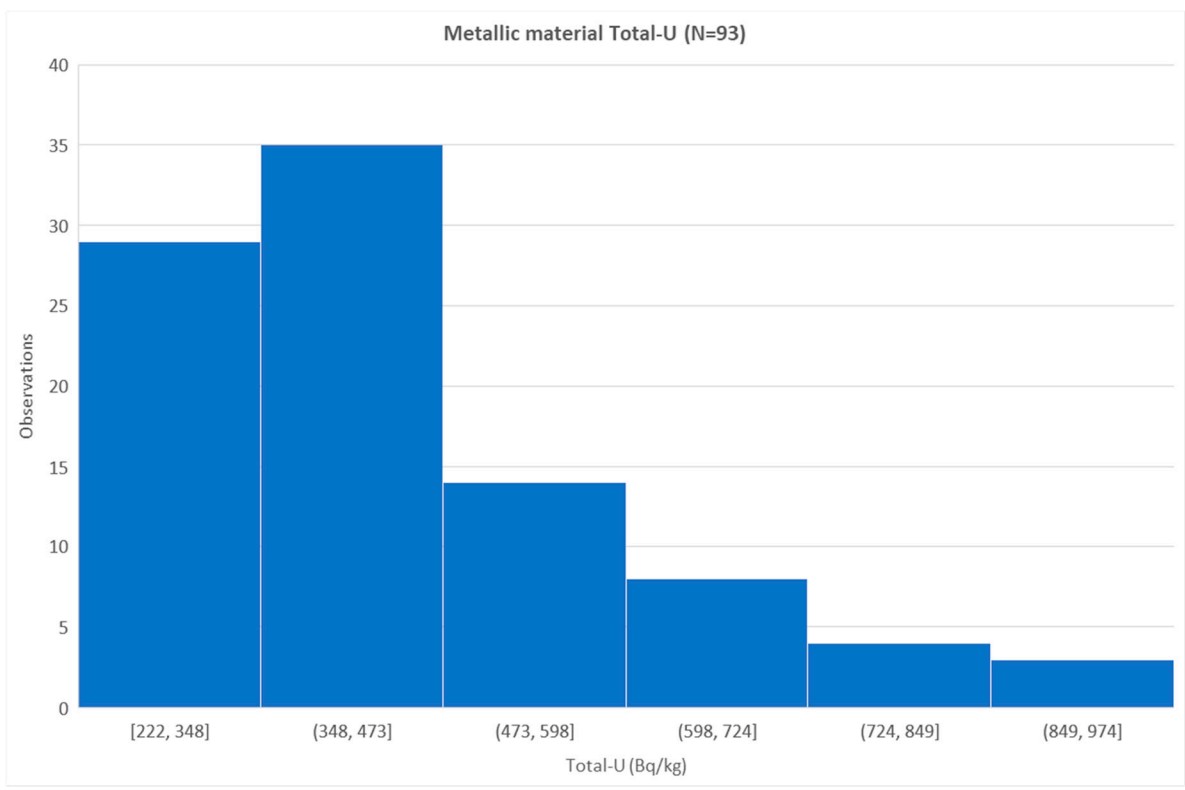

**Figure 3.** Performed radiometric measurements on metallic materials to be cleared from Bosco Marengo "FN" nuclear installation.

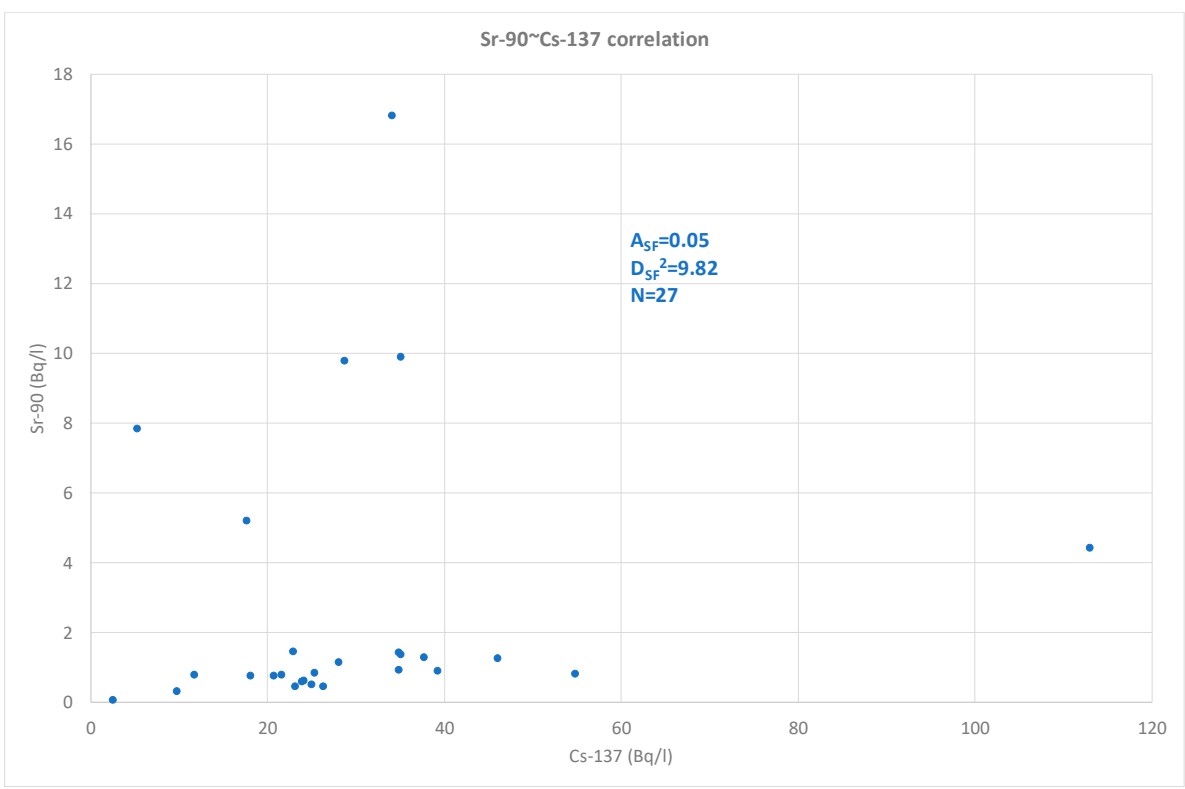

**Figure 4.** Sr-90~Cs-137 correlation in the Trino "E. Fermi" nuclear power plant's liquid effluents.

However, a more detailed analysis—carried out by means of simple sample grouping cycles using R software scripts—made it possible to identify two statistically significant groups (Figure 5),

both with $D_{SF}{}^2 \leq 6$, indicating the presence of at least two different waste streams. The highest point concentration of Cs-137 (~ 110 Bq/l) does not significantly influence the correlation.

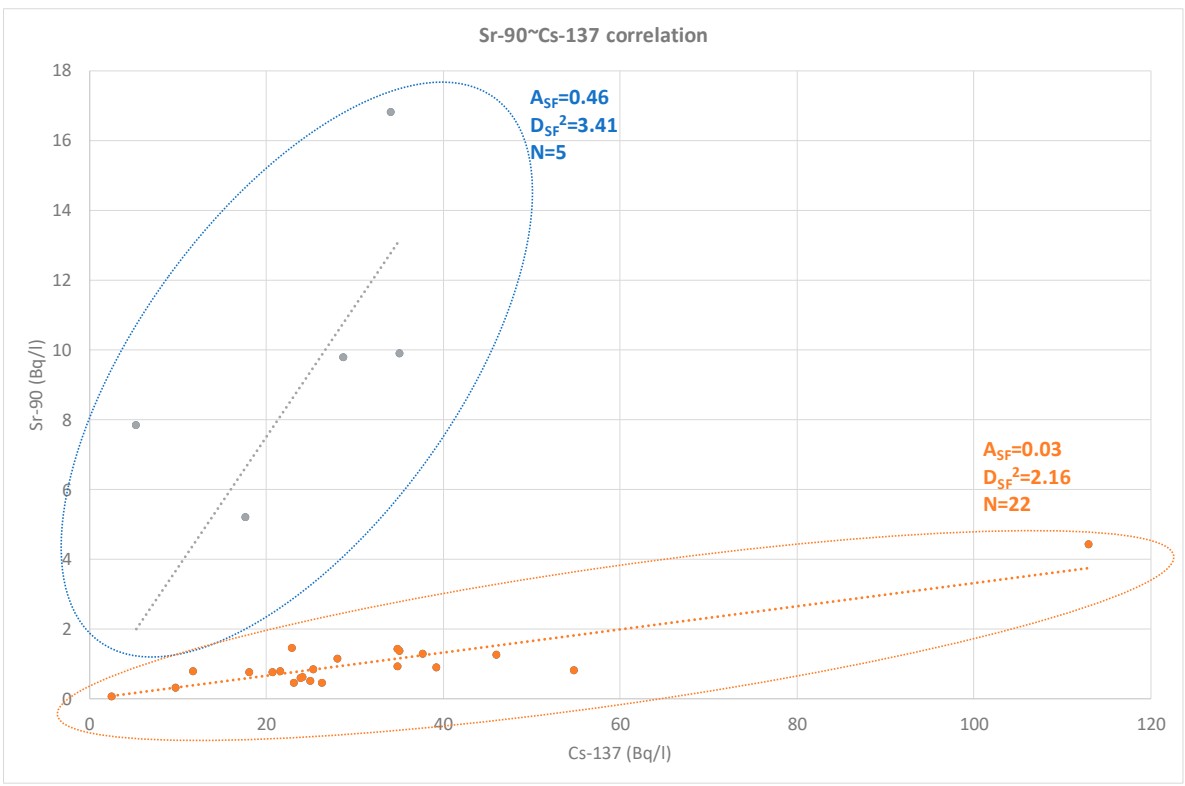

**Figure 5.** Sr-90~Cs-137 correlation in the Trino "E. Fermi" nuclear power plant's liquid effluents—identified groupings have been highlighted.

During 2018, Trino "E. Fermi" nuclear power plant started a treatment activity for part of the potentially clearable radioactive waste, split into three lots.

These wastes were made of material carried out from processing in controlled areas, even though they did not belong to any plant system, classified as radioactive waste at the time of production and that had become clearable over time.

Arpa Piemonte witnessed the initial treatment of some of the samples taken by the operator (10%) and subsequently analyzed the aqueous extracts obtained according to UNI (Ente Nazionale Italiano di Unificazione) standard UNI 11194 [7] in order to make assessments on the scaling factors.

The first analysis of the nature of these samples (sand, cement, earth, metallic material, and clothing) could make one suspect the absence of homogeneity in the composition of the contamination of the waste.

The laboratory analyses, carried out both on ETM and DTM radionuclides, confirmed the absence of homogeneity:

- In wastes with lower contamination (lot 1), the presence of Cs-137, Co-60, and Ni-63 was observed only in half of the samples, and Sr-90 in a single sample. The presence of alpha emitters was not observed.
- In the wastes with high contamination (lot 3), the presence of Cs-137 was observed beyond the clearance levels (Table 4), Co-60, Ni-63, and Sr-90 in only half of the samples, and alpha emitters in only two samples.

**Table 4.** Metallic materials and various materials clearance levels. Technical prescriptions attached to the Ministerial Decree 2 August, 2012 of the decommissioning authorization.

| Radionuclide | Metallic Materials | | | Various Materials |
|---|---|---|---|---|
| | Reuse Surface (Bq/cm$^2$) | Recycle Surface (Bq/cm$^2$) | Reuse/Recycle Mass (Bq/g) | Reuse/Recycle Mass (Bq/g) |
| H-3 | 10,000 | 100,000 | 1 | 1 |
| C-14 | 1000 | 1000 | 1 | 1 |
| Mn-54 | 10 | 10 | 1 | 0.1 |
| Fe-55 | 1000 | 10,000 | 1 | 1 |
| Co-60 | 1 | 10 | 1 | 0.1 |
| Ni-59 | 10,000 | 10,000 | 1 | 1 |
| Ni-63 | 1000 | 10,000 | 1 | 1 |
| Sr-90 | 10 | 10 | 1 | 1 |
| Sb-125 | 10 | 100 | 1 | 1 |
| Cs-134 | 1 | 10 | 0.1 | 0.1 |
| Cs-137 | 10 | 100 | 1 | 1 |
| Eu-152 | 1 | 10 | 1 | 0.1 |
| Eu-154 | 1 | 10 | 1 | 0.1 |
| $\alpha$ emitters | 0.1 | 0.1 | 0.1 | 0.01 |
| Pu-241 | 10 | 10 | 1 | 1 |

In particular, in addition to the impossibility of estimating DTM radionuclides scaling factors in many samples, the EPRI criteria were never respected (Figure 6).

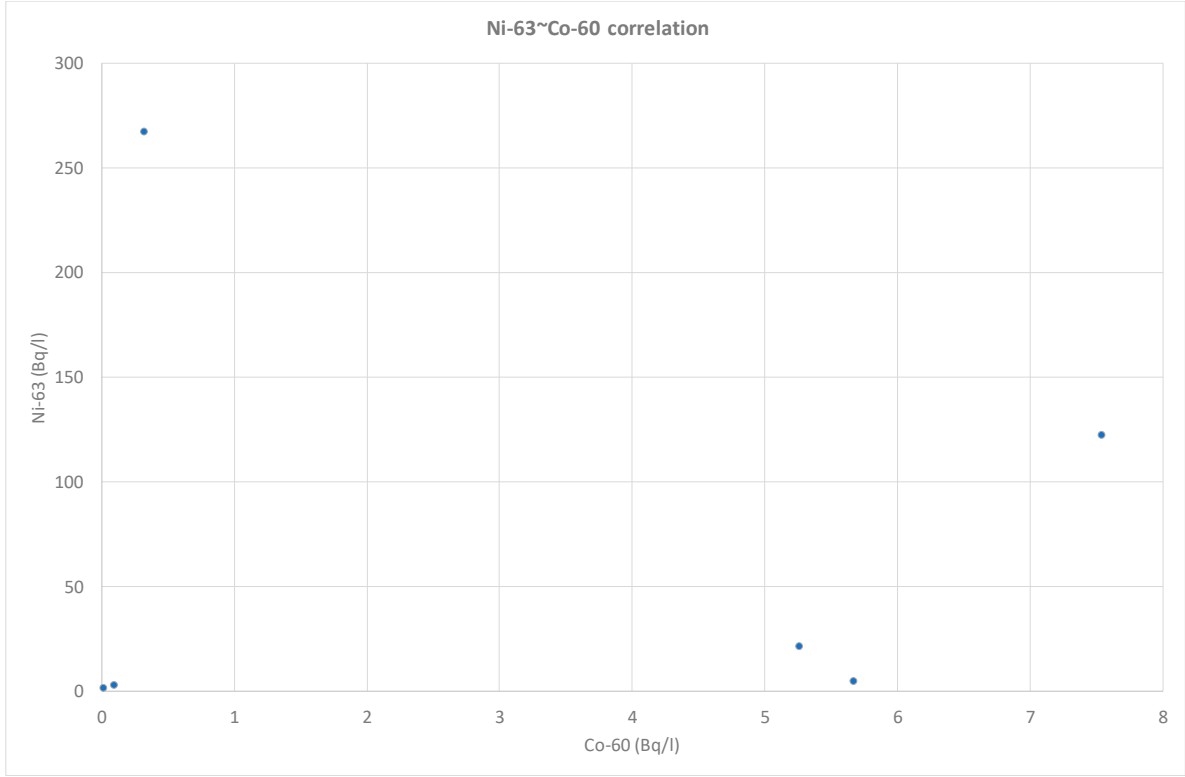

**Figure 6.** Ni-63~Co-60 correlation present in the Trino "E. Fermi" clearable materials.

However, it is possible to highlight the fact that some of the samples belong to one of the two waste streams present in the liquid effluents (Figure 7), confirming that these materials do not form a single homogeneous lot. The condition of homogeneity is a necessary condition for estimating the scaling factors.

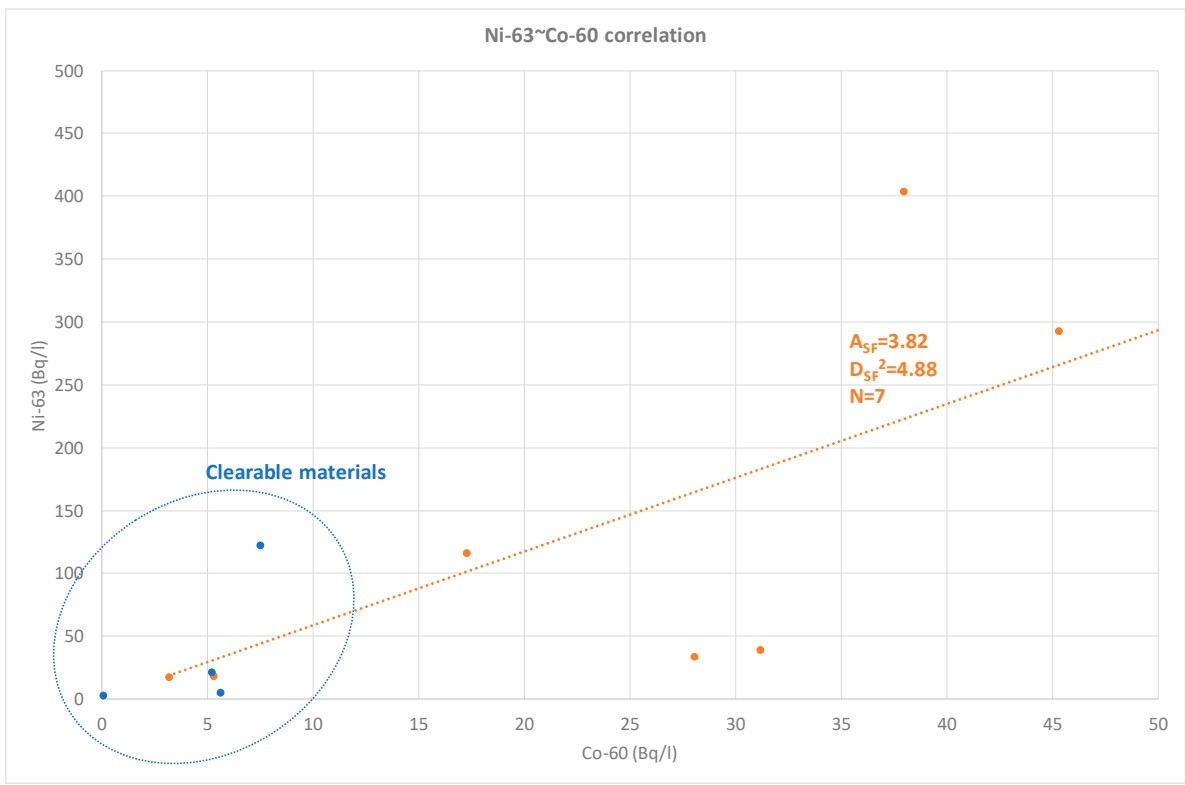

**Figure 7.** Ni-63~Co-60 correlation present in the Trino "E. Fermi" nuclear power plant's liquid effluents, in red, and in some clearable materials samples, in blue.

### 3.3. Saluggia "LivaNova" Nuclear Installation Experiences

At the "LivaNova" nuclear installation based in Saluggia—category A source authorization—for a long time, a hydrocarbon contamination of portions of soil, within the controlled area in which radioactive wastes were also placed, had been highlighted. In this area, during the past years, diesel tanks were placed for supplying an incineration plant, which is now dismantled.

For the purpose of conventional remediation, ISIN has, in any case, requested the radiological characterization of the soil to correctly define its management. In particular, the several possibilities are:

- Soil contaminated by radionuclides (regardless of any hydrocarbon contamination) to be managed as cleared materials or, if the relevant clearance levels are not respected, as radioactive waste.
- Soil without any radiological constraints but contaminated by hydrocarbons to be managed as "special" waste.
- Soil not contaminated by either radionuclides or hydrocarbons.

Soil radiological characterization showed the presence—with considerable inhomogeneities—of the following radionuclides: Cs-137, Sr-90, Am-241, and plutonium. These kinds of radioisotopes had already been identified in 2012 in the contaminated sediments of the radioactive effluent management plant (now decontaminated), in the same ratios. In this case, therefore, by integrating the two datasets, it was possible to preliminarily estimate a complete set of scaling factors characteristic of a single main source of contamination (Figure 8).

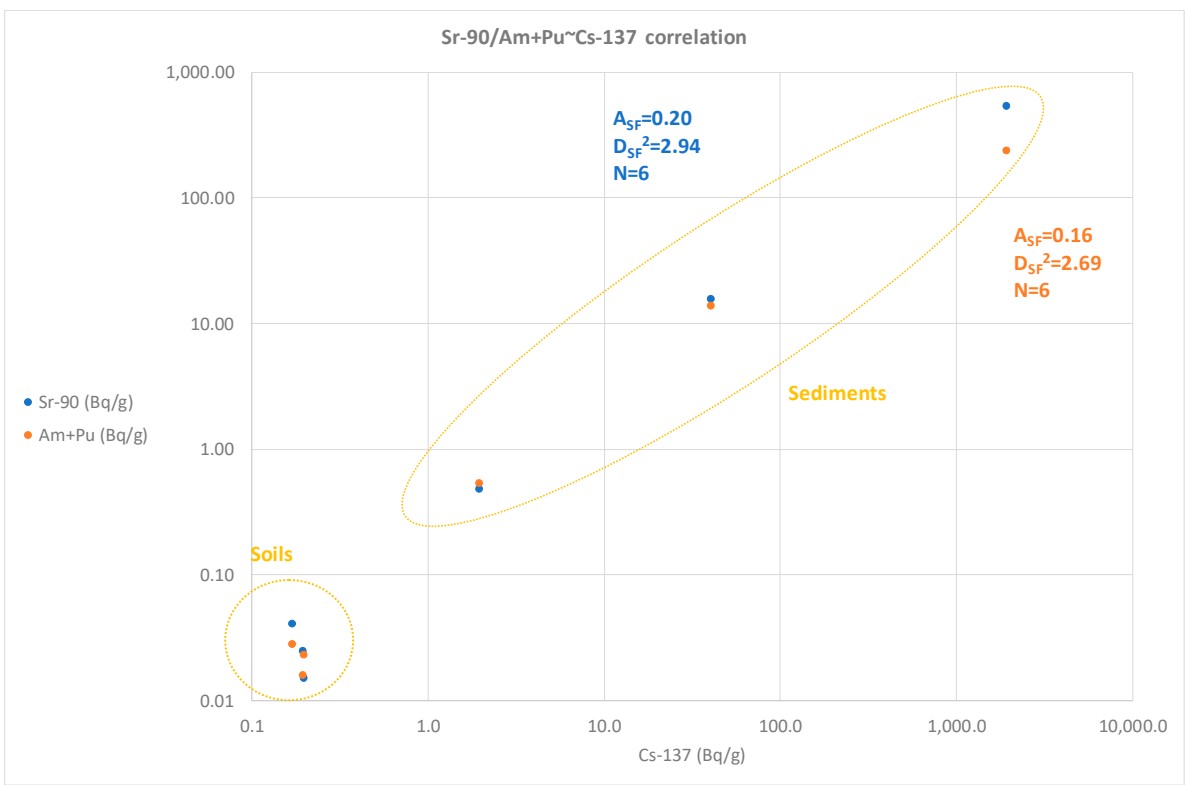

**Figure 8.** Sr-90/Am + Pu~Cs-137 correlation present in the Saluggia "LivaNova" nuclear installation soils and sediments samples.

### 3.4. Saluggia "EUREX" Nuclear Installation Experiences

At the "EUREX" nuclear installation, based in Saluggia—nuclear fuel reprocessing plant—in the year 2017, the radiological contamination of the soil surrounding a portion of the liquid radioactive effluent management plant was highlighted (Figure 9).

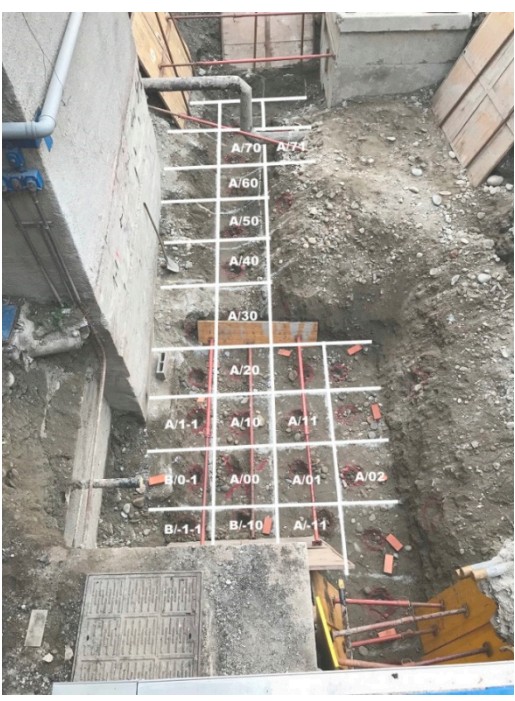

**Figure 9.** Contaminated area.

The correct management of the excavation soil, also in this case, has interested a complete radiological characterization of the soil itself, which showed the presence of only Cs-137. Specifically, however, while for the Sr-90 radionuclide the detection limit was much lower than the relevant clearance level (Table 5)—in particular less than 1%—the detection limits for plutonium isotopes were only in the order of ten times smaller than the corresponding clearance level (Figure 10).

**Table 5.** Cementitious and various materials clearance levels. Ministerial Decree 12 August 2009. Technical management prescriptions.

| Radionuclide | Building Demolition | | Various Materials |
|---|---|---|---|
| | Surface (Bq/cm$^2$) | Mass (Bq/g) | Mass (Bq/g) |
| H-3 | 10,000 | 1 | 1 |
| Ni-59 | 100,000 | 1 | 1 |
| Ni-63 | 100,000 | 1 | 1 |
| Co-60 | 1 | 0.1 | 0.1 |
| Sr-90 | 100 | 1 | 1 |
| Tc-99 | 100 | 1 | 1 |
| Sb-125 | 10 | 1 | 1 |
| Cs-134 | 10 | 0.1 | 0.1 |
| Cs-137 | 10 | 1 | 1 |
| Pm-147 | 10,000 | 1 | 1 |
| Sm-151 | 10,000 | 1 | 1 |
| Eu-152 | 10 | 1 | 0.1 |
| Eu-154 | 10 | 1 | 0.1 |
| Eu-155 | 100 | 1 | 1 |
| α emitters | 1 | 0.1 | 0.01 |
| Other U-238 and U-235 decay products different from Legislative Decree n. 230/1995 Table I-2, Annex I | 0.1 | 0.1 | 0.01 |
| Pu-241 | 100 | 1 | 1 |

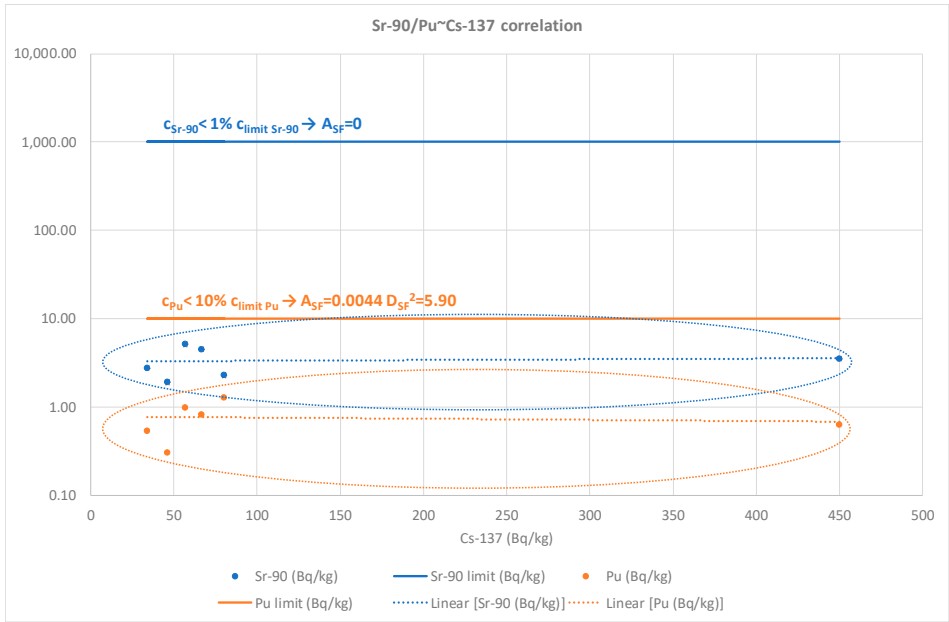

**Figure 10.** Sr-90/Pu~Cs-137 correlation present in the Saluggia "EUREX" nuclear installation soils samples.

If for the Sr-90 it can be considered acceptable setting the scaling factor to zero, for plutonium isotopes, it is more precautionary—also considering the characteristics of the plant—to estimate a non-zero scaling factor, assuming a rectangular distribution between zero and the pertinent detection limits.

## 4. Discussion

Clearance levels' compliance verification should be performed for all radionuclides potentially present, including DTM. It is therefore necessary to carry out upstream, based on a representative number of samples, those radionuclides' determination in order to estimate scaling factors (SF). Specific radiochemistry is used for defining DTMs' concentrations, such as Fe-55, Ni-59, Ni-63, Sr-90, Pu-238, and Pu-239/Pu-240. As a key nuclide, Co-60 was chosen for the activation products (Fe-55, Ni-59, and Ni-63) and Cs-137 for the fission products (Sr-90) and plutonium (Pu- 238, Pu-239/Pu-240, and Pu-241). The presence of very low radioactivity concentrations, often below the detection limits, can make it difficult to determine the related scaling factors.

The different technical experiences carried out so far allows us to formulate the following observations on methods and measurement acceptability performance requirements:

- The homogeneity of a clearable lot of material should always be verified with experimental measurements.
- All available information can be useful for the purposes of estimating scaling factors.
- In all samples used to estimate scaling factors, at least one ETM radionuclide must be detectable.
- It is appropriate to define an exclusion level, for example a detection limit – for the detection limit definition and, more generally, for the definition of the characteristic limits, the reference is represented by the ISO (International Standard Organization) standard ISO 11929 [17] – in the range 1–10% of the relevant clearance level, in order to be able to exclude the presence of DTM radionuclides in the sample.
- It is appropriate to define an acceptability level (for example a detection limit in the range 10–50% of the relevant clearance level) in order to establish whether the sample can be used to estimating scaling factors.

In particular, if radiometric measurements provide for the DTM radionuclide a $C_{DTM}$ activity concentration lower than the corresponding detection limit $DL_{DTM}$, this last detection limit may be prudently adopted for the estimation of the scaling factor (a more cautious assumption from the radiation protection point of view). Alternatively, a rectangular distribution in the range 0–$DL_{DTM}$ could be taken as a reference, adopting $DL_{DTM}/2$ for the purposes of estimating the scaling factor (a less cautious assumption from the radiation protection point of view, perhaps more realistic).

The above criteria, due to preliminary considerations, will have to be evaluated case-by-case, depending on the plant, the specific context, and the specific radionuclide.

In summary, the decisional scheme of Figure 11 may be adopted.

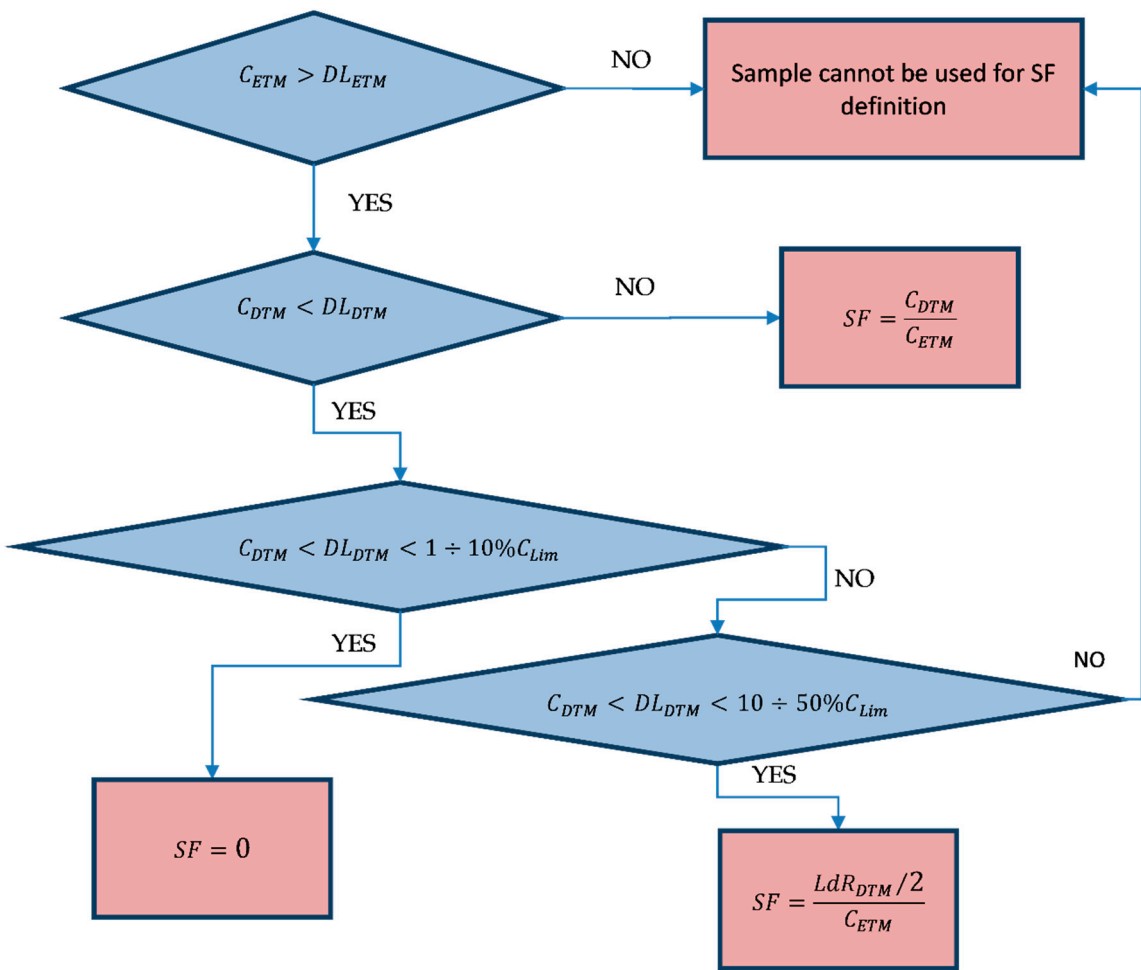

**Figure 11.** Scaling factors' decisional flow chart.

**Funding:** This research received no external funding.

**Conflicts of Interest:** The authors declare no conflict of interest.

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
