# Peer review of "Control Experiences Regarding Clearable Materials from Nuclear Power Plants and Nuclear Installations: Scaling Factors Determination and Measurements’ Acceptance Criteria Definition"

_environments, doi:10.3390/environments6110120_

Round 1
Reviewer 1 Report
The argument, as outlined in the abstract, is of great interest.
On the other hand, the paper full text does not examine in depth all the topics addressed in the abstract especially for what concerns:
measurements methods (gamma spectrometry and radiochemical procedures for different materials and nuclides) methods used for data statistical analysis
I would recommend the Authors to include in the paper more details about the above mentioned topics; I would also suggest to consider the opportunity to discuss in more detail methods used to define and use SFs (logarithmic average and regression, linear / non linear relationship, etc.), in order to highlight the work done to define the approach described in Fig. 11.
Figure 1: not very clear; add legend, if possible modify the axis using a regular scale, move x axis labels
2. Materials and methods: add details or more references, add methods sensitivity
82-84: explain the meaning of 2 sigma dispersion
85-88: add comments on the meaning of DSF2 <= 6
3. Results: a general introduction to the different paragraphs would help the reader to focus on the most relevant topics
Figure 2: add some comments; if possible, add a graph explaining the meaning of ASF and DSF2
Figure 3 and rows 126-127: total U as reported in Figure 3 has been measured by gamma spectrometry? Please check and explain
139-140: add if possible some detail on statistical method used to identify different groups (figure 5) and discuss, add legend to the graph
149-151: explain experimental method used to obtain aqueous extracts, and how they can be considered representative of different types of solid materials
Figure 7: add legend
165-167: not very clear
193: “reports”: correct?
4. Discussion: consider the opportunity to include in the discussion (all?) the topics addressed in the abstract, and/or to focus on the most relevant outcome of the work (working scheme in figure 11? Definition of SFs useful in similar situations? …)
233-237: sentence not clear
Reviewer 2 Report
The paper is correctly written, the introduction and development are clear and the references to official reports on the subject of EPRI or the IAEA are adequate.
The authors have analyzed several cases of nuclear facilities in Italy to obtain a procedure for the determination of the scale factors between various radionuclide groups to obtain an easily measurable representative element.
Based on these analyzes, they propose a methodology to assess whether it is possible to obtain scale factors for a given group of radionuclides and a particular installation.
I think that the paper can be published in its current form, however it would be advisable for the authors to specify the characteristics of the measurement equipment used to obtain the data they use and their limits of detection and uncertainties.
Reviewer 3 Report
Dear Authors,
the presented for review paper titled "Control experiences regarding clearable materials from nuclear power plants and nuclear installations: scaling factors determination and measurements’ acceptance criteria definition" contains the analysis of possible correlations between difficult and easy tomeasure radionuclides in different waste streams and contaminated materials from several selected nuclear instalations in Italy.
I am not really convinced if this paper would be of significant interest for readers of the Environments journal. Paper is quite well written, but in many places it is not easy to follow. Moreover, the problem of prediction of the presence of different difficult to meaure radionuclides in the waste streams is not sufficiently highlighted in the introductory section, which in my opinion should be extended taking into account more literature data. However, apart from above, my special comments are listed below:
1. Line 39-41. "For example, the “constant” ratio between the concentrations of Ce-144 and plutonium isotopes in the waste streams generated by a LWR (Light Water Reactor) was proven." Please provide appropriate reference for this statement.
2. Lines 44-49. Authors write about literature data and problems with their variability, but do not cite any literature. Some examples would be useful. Please add some relevant refernces.
3. Line 60. Figure 1, please provide axes titles and units.
4. Figure 2 and all other graphs. They require modifications, because they are hard to read. Please make the fonts of axes titles and scale numbers larger. In the current form, all these graphs are difficult to be analyzed. Moreover, please replace all values with coma to point as decimal separator.
5.Table 3. "Cementous", should be "cementitious".
6. Figure 3. Missing Y-axis title and unit.
7. Lines 139-140 and Figure 5. Correlation where A_SF=4.6 was obtained seems to be very weak. Could Authors provide some additional explanation for this? Maybe some more information are available for the mixed waste streams? Another comment is related to the second correlation (N=22, A_SF=0.03).Point for Cs-137 activity = ~113 Bq/l may affect in a slightly more significant way to the obtained correlation. Have Authors tried to remove this point from the correlation? How does such operation affect the results?
8. Figure 2, 5 and 7. Please establish the same standard for the given slope in provided equations for the straight line. For example, in the current form in Figure 7 there are 5 significant digits for slope (5.8679), but in Figure 2 only 1 (0.03).
To conclude, in my opinion the presented for review paper cannot be considered as a review article. In fact, Authors present and analyse only few examples of sveral selected nuclear instalations in Italy. Moreover, literature cited is very poor and in my opinion does not allow to classify this paper as a review article. There are numerous publications directly related to the problem of scaling factors used for nuclear waste characterization, as e.g. Determination of scaling factors to estimate the radionuclide inventory of wastes from the IEA-R1 research reactor (Journal of Radioanalytical and Nuclear Chemistry 303(3):2467-2481), Radiological Characterisation from a Waste and Materials End-State Perspective (OECD 2017NEA No. 7373), A Scaling Factor Estimation Program for Low-level Radioactive Waste - 11427 (WM2011 Conference, February 27 - March 3, 2011, Phoenix, AZ) or Nuclear Energy — Nuclear Fuel Technology — The Scaling Factor method to determine the radioactivity of low and intermediate level radioactive waste packages generated at nuclear power plant (International Standard, ISO/DIS 21238, 2005).
Thus, I suggest a major revision for this paper. Manuscript can be accepted after introduction all required corrections, but not as a review article.
Best regards
Round 2
Reviewer 3 Report
The correction done by authors seems to be rushed. Authors did not address all comments raised during revision process. For example, there are no units provided on axes in Figure 1, in other figures still comas are present as decimal separators instead of dots. References [1], [2] and [3] have been cited in a wrong way, it should be [1-3]. I am still not convinced to treat this paper as a review.
I recommend this paper for publication after minor revision.
Author Response
Dear Reviewer
Thank you for your detailed comments.
Effectively, I had little time to review the paper.
I suggest to classify the paper as Case Report - hearing the editor.
Best regards